# Mesospheric bores at southern mid-latitudes observed by ISS-IMAP/VISI; a first report of an undulating wave front

Yuta Hozumi[1], Akinori Saito[2], Takeshi Sakanoi[3], Atsushi Yamazaki[4], and Keisuke Hosokawa[1]

[1]Department of Information and Communication Engineering, University of Electro-Communications, Tokyo, Japan
[2]Graduate School of Science, Kyoto University, Kyoto, Japan
[3]Planetary Plasma and Atmospheric Research Center, Tohoku University, Sendai, Japan
[4]Institute of Space and Astronautical Science, Japan Aerospace Exploration Agency, Sagamihara, Japan

**Correspondence:** Yuta Hozumi (hozumi@uec.ac.jp)

**Abstract.** Large-scale spatial structures of mesospheric bores were observed by the Visible and near Infrared Spectral Imager (VISI) of the ISS-IMAP mission (Ionosphere, Mesosphere, upper Atmosphere and Plasmasphere mapping mission from the International Space Station) in the mesospheric $O_2$ airglow at 762 nm wavelength. Two mesospheric bore events in southern mid-latitudes are reported in this paper; one event at (48° S–54° S and 10° E–20° E) on 9 July 2015, and the other event at (35° S–43° S and 24° W–1° E) on 7 May 2013. For the first event, the temporal evolution of the mesospheric bore was investigated from the difference of two observations in consecutive passes. The estimated eastward speed of the bore is 100 m s$^{-1}$. The number of trailing waves increased with a rate of 3.5 waves h$^{-1}$. Anti-clockwise rotation with a speed of 20° h$^{-1}$ was also recognized. These parameters are similar to those reported by previous studies based on ground-based measurements, and the similarity supports the validity of VISI observation for mesospheric bores. For the second event, VISI captured a mesopshric bore having a large-scale and undulating wave front. The horizontal extent of the wave front was 2,200 km. The long wave front undulated with a wavelength of 1,000 km. The undulating wave front is a new feature of mesospheric bore revealed by the wide field-of-view of VISI. We suggest that non-uniform bore propagating speed due to inhomogeneous background ducting structure might be a cause of the undulation of the wave front. Temperature measurements from the Sounding of the Atmosphere using Broadband Emission Radiometry (SABER) onboard the Thermosphere, Ionosphere, Mesosphere, Energetics and Dynamics (TIMED) satellite indicated that bores of both events were ducted in a temperature inversion layer.

## 1 Introduction

A mesospheric bore is characterized by a propagating, and sharp front in the upper mesosphere. The front is often followed by undulations (undular bore) or turbulence (turbulent bore). Mesospheric bores have been observed as a sharp brightness jump or drop of airglow by ground-based imagers at various longitudes from low latitude to high latitude (e.g., Taylor et al., 1995; Smith et al., 2003, 2017; Fechine et al., 2005; Nielsen et al., 2006; Li et al., 2013; Medeiros et al., 2018).

Dewan and Picard (1998) proposed an explanation of mesospheric bore as a propagating discontinuity in a stable layer or a duct in varicose mode. In varicose mode, the upper and lower surfaces of the layer oscillate symmetrically about the mid-plane. Medeiros et al. (2005) demonstrated that most of bores showed that complementary effect suggested by Dewan and Picard

(1998) in airglow responses, and validated the varicose mode oscillation. By solving conservation of mass and momentum equations, Dewan and Picard (1998) calculated the speed of a bore as follows:

$$U = \sqrt{g' \frac{h_1(h_1 + h_0)}{2h_0}} \tag{1}$$

Here, $h_0$ and $h_1$ are half width of a ducting layer upward and downward of a bore, respectively. $g'$ is the gravitational accel-
eration corrected for buoyancy force. The mesospheric bore model in a thermal duct has been examined and validated using simultaneous lidars and airglow imagers observations (Smith et al., 2003, 2005; She et al., 2004). It has been also demonstrated with simultaneous radar observations that wind has an important role for background condition of mesospheric bore (Fechine et al., 2009; Giongo et al., 2018). Dewan and Picard (2001) provided a possible explanation of mesospheric bores through critical layer interaction of gravity waves with the mean flow. Seyler (2005) and Laughman et al. (2009) demonstrated,
by using a numerical simulation, that a wave front of a long-wavelength gravity wave in a duct can steepen and become a bore-like sharp front. Such a formation of a mesospheric bore from a large-scale gravity waves was reported by Yue et al. (2010) utilizing ground-based observations. However, the generation mechanism of mesospheric bores are still not fully understood.

Historically, the study of mesospheric bores have been conducted based on ground-based measurements. Recently, Miller et al. (2015) reported two space-borne observations of mesospheric bore events by Day/Night Band (DNB) onboard the NOAA/NASA
Suomi National Polar-orbiting Partnership environmental satellite. Although there are successful observations of mesospheric bores by DNB, a lot of works are left to do on the mesospheric bore study with space-borne imaging. First, the geographical variation on the bore characteristics is not fully understood. The number of ground-based observation site is limited, and the locations of observation site are restricted by land-sea distribution. There are reports of bore observations at low latitudes (e.g., Medeiros et al., 2001; Fechine et al., 2005; Medeiros et al., 2018), northern mid-latitudes (e.g., Taylor et al., 1995; Smith et al.,
2003, 2017; Li et al., 2013), and southern high-latitude (e.g., Nielsen et al., 2006; Li et al., 2013; Medeiros et al., 2018), but there is no reports in southern mid-latitudes and northern high-latitudes. Second, the large-scale horizontal structure of bores is unclear. Since ground-based imagers have often observed only a portion of bore's wave front, the typical horizontal spatial scale of mesospheric bore seems to be larger than the imagers' field-of-view (FOV). Previous studies based on ground-based observations have provided limited information on bore's large-scale horizontal structure. Space-borne airglow imaging is a
strong tool to study mesospheric bores with global observational coverage and a wider FOV, and can overcome the limitations of ground-based observations.

Visible and near Infrared Spectral Imager (VISI) of the Ionosphere, Mesosphere, upper Atmosphere and Plasmasphere mapping mission from the International Space Station (ISS-IMAP mission) is another instrument that has a capability to image mesospheric airglow from space with a wide FOV. While Miller et al. (2015) limited their focus to illustrate the DNB's potential
for bore observation, we report two successful bore events from VISI with further detailed analyses to address the two topic, the latitudinal difference of the bore characteristics, and the large horizontal structure. In this paper, we report two mesospheric bores observed by VISI at southern mid-latitudes. The bore of event #1 was captured in two consecutive passes by VISI, thus, the temporal evolution of the structure can be investigated from the difference of two images. The bore showed counter clockwise rotation, while previous studies report clockwise rotation of bore front at northern mid-latitudes (Smith et al., 2003;

Li et al., 2013). As event #2, we report a mesospheric bore having a very long wave front exceeding 2,200 km. With a benefit of VISI's wide FOV, a new feature of horizontally undulating wave front was captured. The vertical undulations following bore fronts have been often discussed since Dewan and Picard (1998) explained the following undulation with vertical displacement in varicose mode. However, horizontal undulation of mesospheric bore front has never been reported as far as we know. This paper is the first report of an undulating bore front.

## 2 Instrumentation and Methodology

VISI is a visible and near infrared spectral imager that was installed on the International Space Station (ISS), and made observations from September 2012 to August 2015. VISI has a grism as a disperser, and the spectral resolution $(\lambda/\Delta\lambda)$ is $\sim$800. The detector is a back-illuminated, frame transfer CCD (e2V 47-20 AIMO) with $1024 \times 1024$ pixel format, $13 \times 13$ $\mu$m single pixel size, and 0.92 of quantum efficiency at 630 nm. The column axis of CCD is for space, and the raw axis is for wavelength. On-chip binning is performed along the column (spatial) axis with a nominal binning size of 16 pixel. It has two slit-shape FOVs that are perpendicular to the ISS orbit track and 45° forward/backward to nadir. VISI performs continuous line-scan imaging and provides seamless two-dimensional image of airglows. The nominal exposure time is 1.0 s. Since it requires 0.9 s reading time, the exposure cycle is 1.9 s. The main targets of VISI are $O_2$ (0-0) at 762 nm, OH Meinel at 730 nm, and OI at 630 nm. In the nominal operation, the peak(maximum) and background (minimum) counts in the wavelength range around the center (762 nm, 730 nm or 630 nm) $\pm$ 6 nm are recorded. By subtracting the background count from the peak count, and multiplying the calibration factor, the airglow intensity is obtained. See Sakanoi et al. (2011) for more detail of the instrumentation.

Data of $O_2$ (0-0) airglow at 762 nm was utilized in this study. Because there is a strong absorption by the ground state $O_2$ in the lower atmosphere, the $O_2$(0-0) band is not sensitive to contamination from lower altitude, such as city lights or the moon light refraction from cloud top. The nominal altitude of the emission is 95 km so that data were mapped to the altitude of 95 km. Assuming an ellipsoid with the eccentricity of World Geodetic System 84 (WGS84) and an equatorial axis of wgs84A (6378.137 km) + 95 km as the altitude plane of 95 km, the intersection point between the ellipsoid surface and the line-of-sight of each pixel was calculated. Then, we mapped the airglow intensity to the intersection point, and obtained the two-dimensional airglow image. At the mapping altitude, the horizontal width of FOV is 670 km. The spatial resolutions are 13 km along and 12-15 km across the ISS orbit track. The typical phase speed of mesospheric bores is in the range of 60–80 m s$^{-1}$ (e.g., Taylor et al., 1995; Medeiros et al., 2001; Smith et al., 2003; She et al., 2004; Smith et al., 2005; Nielsen et al., 2006; Narayanan et al., 2009; Fechine et al., 2009; Bageston et al., 2011; Li et al., 2013; Giongo et al., 2018). To our knowledge, the highest bore phase speeds reported in the literature are $98 \pm 8$ m s$^{-1}$ (Brown et al., 2004). The speed of the ISS is 7.4 km s$^{-1}$, which is significantly higher than the typical phase speed of mesospheric bores. Therefore, it is reasonable to consider that VISI observes snapshots of bore induced airglow structure.

Temperature profiles obtained by the Sounding of the Atmosphere using Broadband Emission Radiometry (SABER) instrument onboard the Thermosphere, Ionosphere, Mesosphere, Energetics and Dynamics (TIMED) satellite were employed

as supporting data to assess the background thermal structure. Vertical profile of kinetic temperature can be retrieved from SABER measurements of $CO_2$ 1.5 $\mu$m earth rim emission (Mertens et al., 2001). The v2.0 level 2A data downloaded from http://saber.gats-inc.com was used in this study. The square Brunt-Väisälä frequency can be derived from a temperature profile as

$$N^2 = \frac{g}{T}\left(\Delta T_z + \frac{g}{C_p}\right) \tag{2}$$

where $T$ is the temperature, $\Delta T_z$ is the vertical temperature gradient, and $C_p$ is the specific heat at constant pressure. The square Brunt-Väisälä frequency at an altitude of $z$ is obtained from two continuous SABER temperature data by substituting $T = (T(z_1) + T(z_2))/2$ and $\Delta T_z = (T(z_2) - T(z_2))/(z_2 - z_1)$ into equation (2) where $z = (z_1 + z_2)/2$. The height step of the SABER data around 95 km is $\sim 0.4$ km.

## 3   Results and Discussion

### 3.1   Event #1

A sharp front followed by undulations was observed by VISI on 9 July 2015 over the south Atlantic ocean. The front was captured in two consecutive passes so that the temporal evolution of the structure can be investigated from the observations. The airglow images obtained by the forward and backward FOVs of VISI are shown in Figure 1. In the first pass, the center of the forward and backward FOVs crossed the front at 19:11:13 and 19:12:52 UT. In the second pass, they crossed it again at 20:49:13 UT and 20:50:52 UT, respectively.

In the first observations (Figure 1a, d), a sharp brightness jump of airglow followed by wave structures can be seen around 10° E longitude with S-N elongation of the front. The western (eastern) side of the front is bright (dark), and the boundary is fine. We can see small wave trains whose wavefronts parallel to the front of brightness jump on the western side. The front and wave crests identified in VISI images are indicated with blue lines in Figure 1e, f. The morphological feature is exactly the same as bore induced airglow structures reported by previous studies. Since the undulations are seen on the western side of the front, the mesospheric bore is expected to propagate eastward. Two crests were identified on the southern part of the western side of the front. The interval of the crests is $\sim$30 km. Then, the wavelength of following waves is $\sim$30 km. The $O_2$ airglow brightness on the western side of the front is 2,500–3,000 R, which is 250–300 % brighter than that on the eastern side. Another enhancement of airglow brightness can be seen at 15° E–18° E, but the boundary of this enhancement is not sharp, and we will not focus on this enhancement in this study.

In the second observations (Figure 1b,d), the front moved eastward, and was observed at 16° E–25° E. The amplitude of the brightness jump is almost same as that in the first pass, 3,000–3,500 R in the western (bright) side as compared to 1,300 R in the eastern (dark) side. The airglow intensities both sides of the wave front were slightly brighter than the intensities in the previous pass. At 50° S, the front moved 620 km eastward in the interval. Thus, if we assume a pure eastward propagation, the bore speed is estimated as 100 m s$^{-1}$. This value is close to the bore speeds reported by previous studies that are typically 60–80 m s$^{-1}$. The undulations were identified on the southern part in the second pass as well as in the first pass. As shown in

the Figure 1e, f, the number of wave crests increased to seven in the second pass from two in the first pass. It means that wave adding rate is expected to be 3.5 waves h$^{-1}$. This value is also similar to the wave adding rates reported previously that ranged from 1.3 to 8.6 waves h$^{-1}$ (Taylor et al., 1995; Smith et al., 2003; She et al., 2004; Nielsen et al., 2006; Narayanan et al., 2009; Li et al., 2013; Smith et al., 2017).

Mesospheric bores are thought to require a channel or region of increased stability, in which a bore propagates (Dewan and Picard, 1998). A mesospheric inversion layer (MIL), a large wind shear, or the combination of both can make such a structure. To assess the background condition of this bore event, both temperature and wind data are ideally needed. For the current case, however, wind data from radar or lidar is not available. Thus, we examine only background temperature structure with TIMED/SABER data. TIMED/SABER made a near-coincident observation (orbit 73617, event 82) 2.3 hour after the second observation of

VISI, at 23:10 UT on 9 July 2015, at (52.1° S, 24.8° E) for 95 km altitude (Figure 2a). Figure 2b presents the vertical temperature profile showing a 30 K temperature inversion at 95–100 km altitude, and a 10 k temperature inversion at 91–93 km. The vertical profile of Brunt-Väisälä Frequency derived from SABER temperature measurement is presented in Figure 2c. A stable layer, the region of strong stability (high $N^2$) surrounded by low stability or unstable regions (low or negative $N^2$), is recognized at 95-100 km altitude. This is a favorable structure for mesospheric bores in which a bore propagate in varicose

mode. An unstable region around 95 km appears narrow, while an unstable region at 90–91 km is relatively thick. Then, the lower limit of the stable region might be 91 km. In any case, TIMED/SABER data shows the existence of a stable layer near the typical emission peak altitude (95 km) of 762 nm $O_2$ emission. Since temperature inversions are thought to be long-lived phenomena (Meriwether and Gardner, 2000; Meriwether and Gerrard, 2004), the stable layer due to the inversion layer observed by SABER is expected to exist during the VISI observation, 2.3 hours before the SABER observation. The mesospheric bore

is likely to propagate in the stable layer created by a temperature inversion.

The orientation of the wave front was NNW-SSE direction in the second pass, while it was almost N-S direction in the first pass. The azimuthal angle of the wave front was observed to rotate counter clockwise from 12° (angle from north to east) to -18° within one orbital period of the ISS. Thus, the rotation speed is -20° h$^{-1}$. The difference of the phase speed of the front in the southern side and northern side might be a cause of the rotation. According to equation (1), propagating speed of bore

depends on the depth of the ducting layer ($U \propto \sqrt{\text{duct depth}}$) and surrounded temperature structure (via $g'$). If the layer depth is thicker at the southern side, the phase speed is larger there, then, the front rotates counter clockwise.

The background wind would be another important factor when we consider the rotation of the front. Generally, wind in the mesopause region is largely dominated by atmospheric tides and inertia gravity waves, and a majority of those are thought to propagate from lower atmosphere (e.g., Aso and Vincent, 1982; Nakamura et al., 1993). In winter mid-latitude, the semi-

diurnal and diurnal tides have significant amplitude (Aso and Vincent, 1982). Upward propagating tides and waves make clockwise variations in the northern hemisphere and counter clockwise variations in the southern hemisphere due to the Coriolis acceleration. While the current bore in the southern mid-latitude showed counter clockwise rotation, northern mid-latitude bores reported in previous studies showed clockwise rotation. Smith et al. (2003) reported 6° h$^{-1}$ clockwise rotation of a bore front over North America. Li et al. (2013) reported 8° h$^{-1}$ clockwise rotation over northern China. Rotation directions of these

bores in both northern and southern hemisphere are consistent with the expected background wind variations that dominated

by upward propagating tides and waves. The rotation of wave fronts might be caused by background wind variations. In the past studies, to our knowledge, there was no report of mesospheric bore at southern mid-latitudes, where the ground-based observation sites are not dense. VISI can provide the southern mid-latitude data, and show the hemispheric difference clearly.

The bore parameters, such as phase speed, wavelength of trailing waves, and wave adding rate, were similar to those reported by previous studies based on ground-based measurements. These similarities support the validity of VISI observation for mesospheric bores. Next, we will report a large-scale spatial structure of mesospheric bore revealed by the wide FOV of VISI.

## 3.2   Event #2

A mesospheric bore having a very long wave front was observed on 7 May 2013 over the south Atlantic Ocean ($35°$ S–$43°$ S, $24°$ W–$1°$ E) as shown in Figure 3. A front, characterized by a sharp increase in brightness, elongated W to NW and E to SE was captured from 20:55:01 UT to 20:59:57 UT with the forward FOV, and from 20:55:51 UT to 21:01:26 UT with the backward FOV. The $O_2$ airglow brightness on the SSW side of the front is 4,200–5,000 R, that is 130-160 % brighter than that on the NNE side. Within the VISI's FOV, the wave front has a large horizontal extent exceeding 2,200 km.The small wavy structures parallel to the wave front are seen at $8°$ W–$0°$ W on the SSW (bright) side of the wave front, thus, the bright front is expected to propagate NNE-ward. The wavelength of trailing waves is 50 km. An interesting feature of this event is the undulating wave front. The wave front of the bright jump was not straight. It undulated with a wavelength of $\sim$1,000 km. The crests of the modulation are at ($18°$ W, $27°$ S) and ($6°$ W, $40°$ S), where the front advanced forward to the propagating direction (NNE) compared to other portions of the front.

As same as event #1, there is no available wind data for this event. We examine only background temperature structure with TIMED/SABER data. Figure 4 shows the result of SABER measurement 1.5–4.5 hours after the VISI observation. Since the horizontal extent of the bore front is zonally long, temperature data from three passes are presented. The nearest measurement to the bore front was made at ($25°$ E, $40°$ S) for 95 km altitude at 23:10 UT on 7 May 2013 (Obrbit 61836, event 57, blue line). In the temperature profile, a large 60 K temperature inversion layer is recognized at 93–100 km altitude. Figure 2c indicates the existence of corresponding stable layer at 93–97 km favorable for mesospheric bore propagation. In the temperature profile of the previous pass (orbit 61835, event 58, red line), a temperature inversion can be seen at the similar altitude (93–95 km), but the intensity of inversion is much smaller. It is less than 30 K. The temperature profile of the following pass (orbit 61837, event 55, green line) shows temperature inversions at 89–93 km and 100–105 km, they are a little lower and higher than previous two passes. Since three temperature profiles show an existence of temperature inversion layer near the emission height, the mesospheric bore of event #2 seems to have propagated in the stable layer created by a temperature inversion as same as event #1.

In this event, VISI captured a very large spatial extent of mesospheric bore exceeding 2,200 km long. The result indicates that the propagating region due to a mesospheric inversion layer has similar or larger spatial extent. Although the three SABER temperature profiles obtained with 1,000 km interval in horizontal and 1.5 hour time interval show small discrepancies in the inversion amplitude and layer altitude, the stable layer caused by the inversion layer is expected to have one continuous large horizontal extension at the time when bore propagated. Smith et al. (2003) reported a mesospheric bore propagating for 1,000

km and a concurrent measurement of an inversion layer at a separated observation site, and showed that inversion layers with horizontal scale-sizes of 1,000–1,500 km can exist. Suzuki et al. (2013) demonstrated that ducting gravity wave can propagate over 1,800 km at mesopause height by using a network of ground-based imagers. These previous studies showed the horizontal extents of propagating or ducting medium along the propagating direction of waves. Our result shows that the horizontal extents

can be also large perpendicular to the propagating direction as it exceeds 2,200 km.

While a curved or bent wave front of a mesospheric bore has been reported (Brown et al., 2004; Smith et al., 2017), to our knowledge, an undulating wave front of mesospheric bore has never been reported. The undulating wave front seen in Figure 3 is a new feature of mesospheric bore that was revealed by the wide FOV of space-borne imaging. A possible explanation of the undulated wave front is non-uniform bore speed along the wave front. As illustrated in Figure 5a, periodically modulating

propagating speed can make an undulating wave front. As mentioned above, propagating speed of bore front depends on the depth of the ducting layer and surrounded temperature structure. A spatial inhomogeneity in the ducting structure, such as duct depth or thermal structure, can cause an inhomogeneity of bore speed in space. Figure 5b shows a schematic picture of the possible modulating duct structure. Since bore speed is proportional to the square root of duct depth, larger bore speed is expected where duct depth is large, and smaller bore speed is expected where duct depth is small. Thus, if there is a

modulating duct as shown in Figure 5b, it can yield non-uniform bore speed, and resultant modulating wave front. The bore front modulation in Figure 3 shows sinusoidal shape. It implies that the spatial inhomogeneity in the duct depth or temperature gradient has also sinusoidal modulation. This fact suggests that the modulation in the duct depth or temperature gradient might be a result of the interaction between the stable layer and a wave with horizontal wavelength of 1,000 km.

A point-like tropospheric source location of an atmospheric gravity wave can be found by estimating the curvature of the

observed wave front from in airglow imagery with the assumption that the disturbance propagates radially from its source (Suzuki et al., 2007). An implication from the discussion of the undulated wave front is that such a method for source identification using a curvature of wave front would be misleading in case the bore speed is non-uniform.

## 4  Summary

Two mesospheric bore events at southern mid-latitudes observed by VISI were reported. For event #1, the temporal evolution

of bore was estimated from two consecutive VISI observations. Estimated bore parameters, such as phase speed, wavelength of trailing waves, and wave adding rate, were consistent with previous studies. These results validates the use of VISI for bore studies. This event provided information on the temporal evolution of the azimuth angle of bore front in the southern mid-latitudes. The bore was rotated counter clockwise with a speed of $20° \text{ h}^{-1}$, while past studies reported clockwise rotating bore in the northern mid-latitudes. The rotating directions are consistent with upward propagating tides and gravity waves

suggesting that the bore was affected by tidal backward wind variation. From event #2, with the benefit of the wide FOV of VISI, we found that 1) a wave front of a mesospheric bore can be long as it exceeds 2,200 km, 2) a wave front of a mesospheric bore is not always straight or simply curved in a large-scale view, it can periodically undulate. The undulating wave front of bore suggests that the bore speed in the duct is not uniform in space. The space-borne imaging has a wide FOV and global

observation coverage, and it can be utilized to study mesospheric bore in synoptic scale. As a future work, we plan to conduct a statistical study on the global characteristic of mesospheric bore with VISI data.

*Data availability.* ISS-IMAP/VISI data are available via e-mail inquiry to Akinori Saito at Kyoto University (saitoua@kugi.kyoto-u.ac.jp).

5    TIMED/SABER v2.0 level 2A data were downloaded from http://saber.gats-inc.com.

*Competing interests.* The authors declare that they have no conflict of interest.

*Acknowledgements.* Data utilized in this study are from the Ionosphere, Mesosphere, upper Atmosphere, and Plasmasphere mapping mission from the ISS (ISS-IMAP mission). We thank all the member of the ISS-IMAP mission. We also acknowledge SABER team for the SABER temperature data. We thank three anonymous referees for their comments on the open discussion.

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

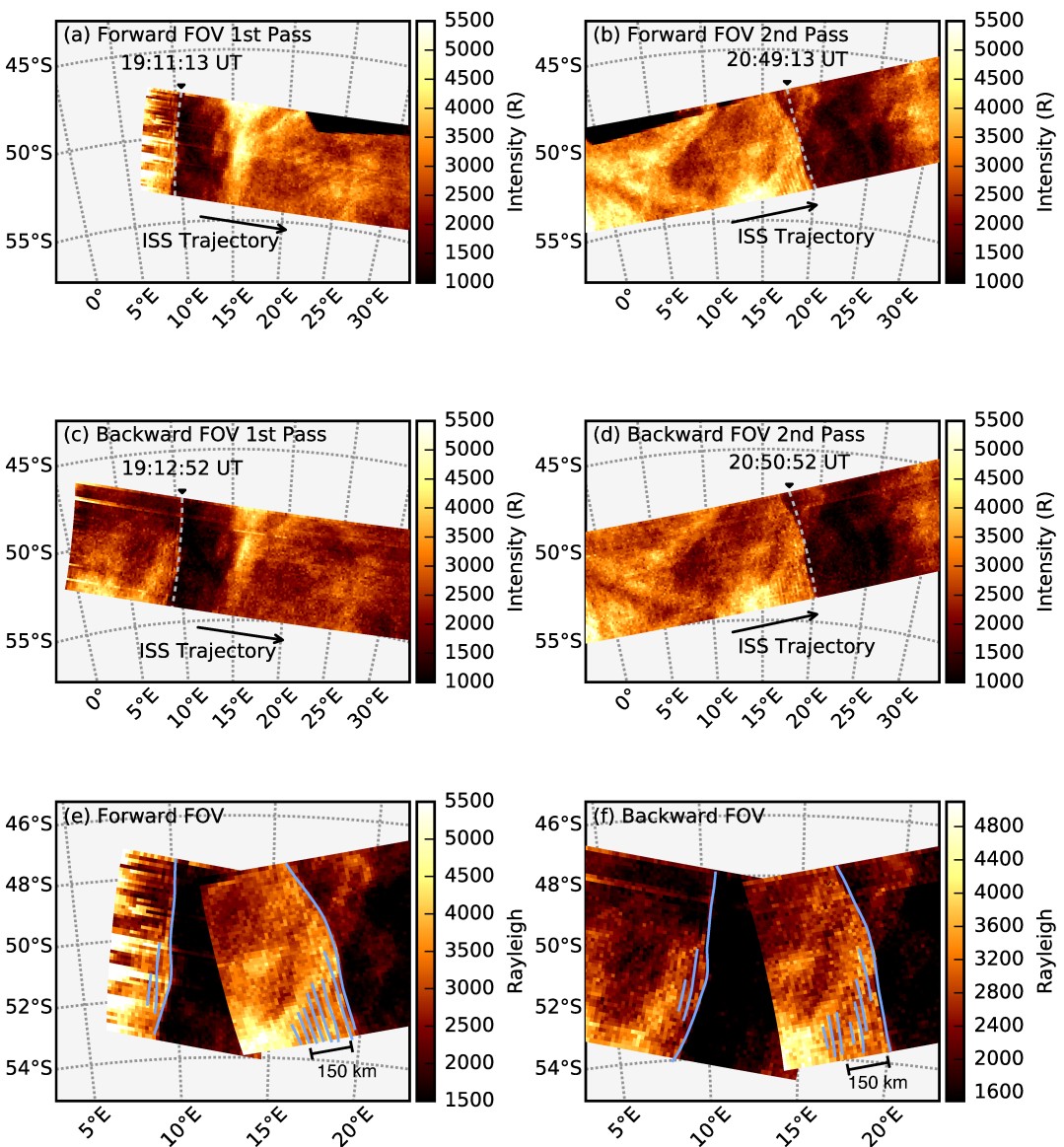

**Figure 1.** $O_2$ (762 nm) airglow images obtained by VISI with the forward and backward FOVs in two consecutive passes on 9 July 2015. Airglow images were mapped to the altitude of 95 km. (a), (b), the mesospheric bore observed by the forward FOV in the first pass and second pass, respectively. The FOV crossed the front at 19:11:13 UT and 20:49:13 UT. (c), (d), the mesospheric bore observed by the backward FOV in the first pass and second pass, respectively. The FOV crossed the front at 19:12:52 UT and 20:50:52 UT. (e), (f), the airglow images obtained in the two passes are shown in one plot. The airglow data are same as above 4 plots, but the images are zoomed in to show detailed wave characteristics. Blue lines indicate the bore front and the wave crests.

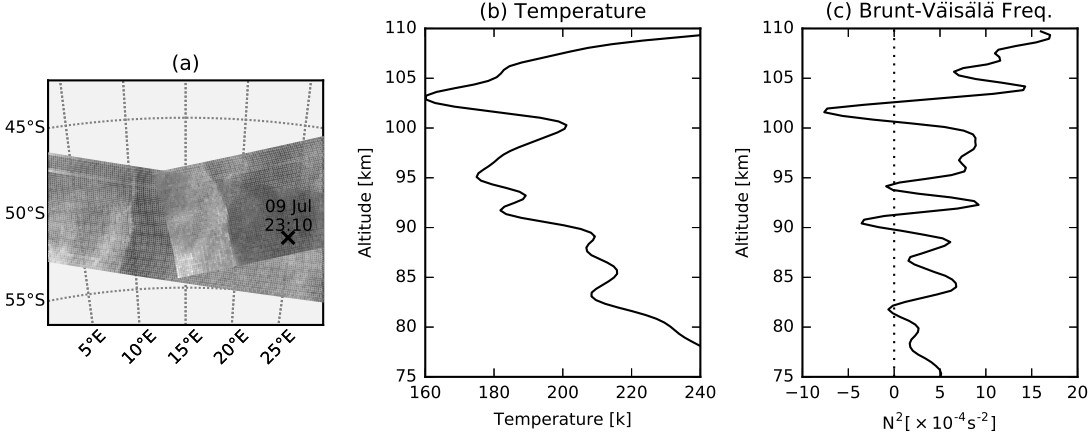

**Figure 2.** SABER measurements obtained 2.3 hours after the bore event #1. (a) The measurement point at 95 km altitude is indicated by black cross. (b) The altitude profile of temperature showing a 30 K temperature inversion at 95-100 km altitude. (c) The altitude profile of resulting Brunt-Väisälä frequency showing a stable layer at 95-100 km.

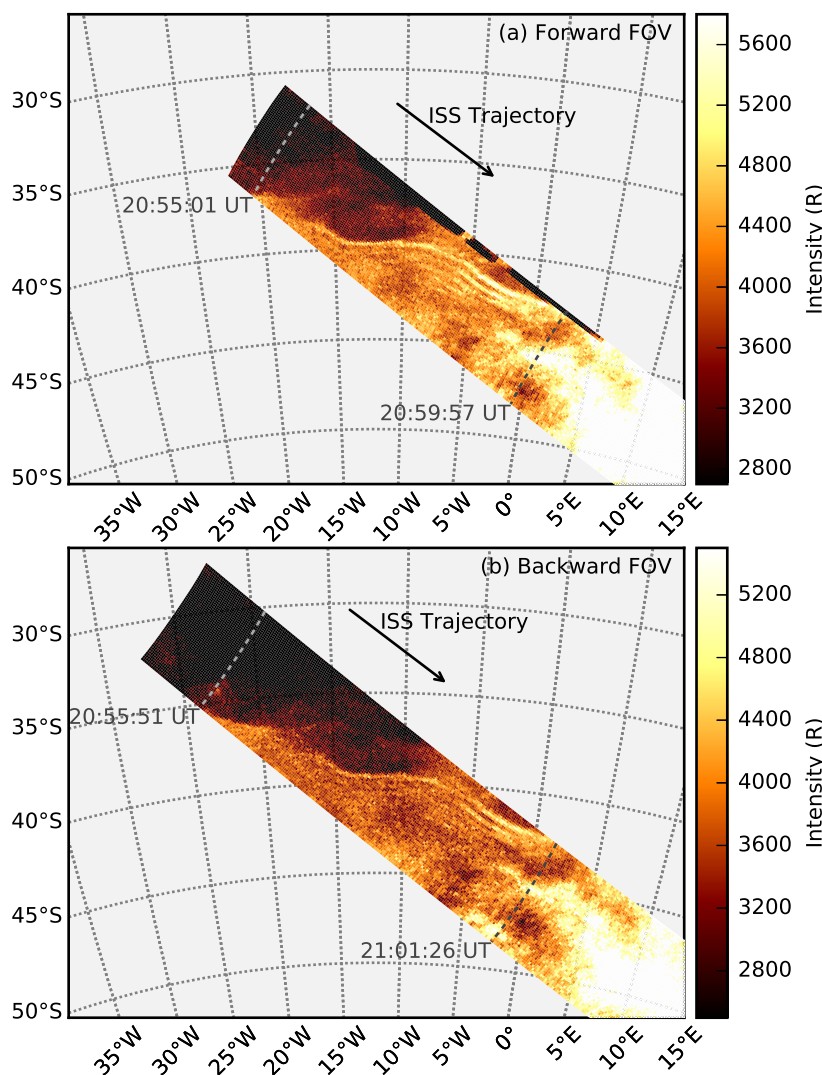

**Figure 3.** O$_2$ (762 nm) airglow images obtained by VISI with the forward and backward FOV on 7 May 2013. The start and end times of capturing the bore front are indicated with dashed lines.

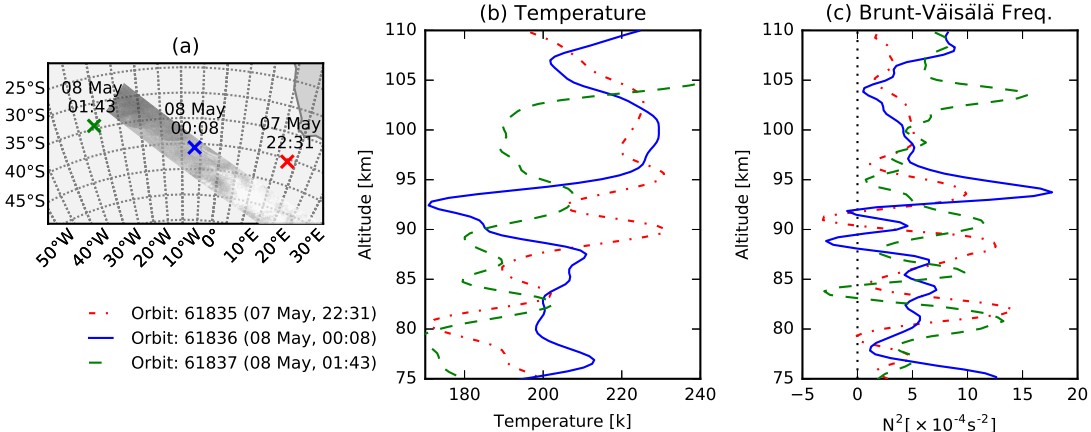

**Figure 4.** SABER measurements obtained from 7 May 22:31 UT to 8 May 01:43 UT. (a) The measurement points at 95 km are indicated by crosses. (b) The altitude profiles of temperature. (c) The altitude profiles of Brunt-Väisälä frequency.

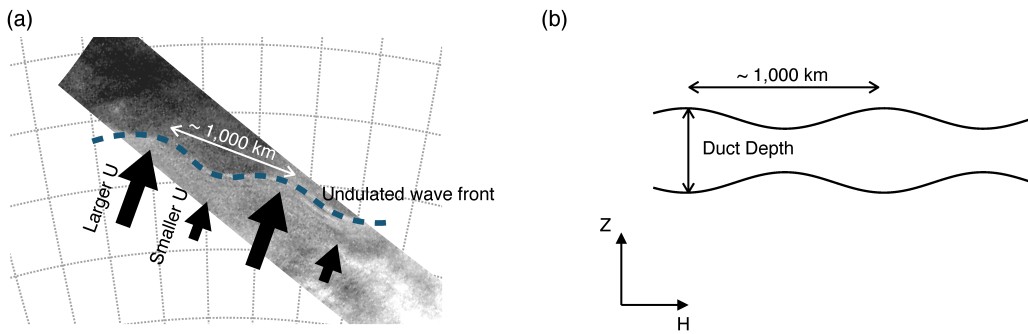

**Figure 5.** (a) Possible interpretation of the modulating wave front. (b) Schematic picture of a proposed modulating duct structure.