# Peer review of "Mesospheric bores at southern mid-latitudes observed by ISS-IMAP/VISI; a first report of an undulating wave front"

_Atmospheric Chemistry and Physics, 2018_

## Referee Comment (RC1) · Anonymous Referee #1 · 4 Jun 2018

This paper reports two mesospheric bores in the southern hemisphere observed by ISS-IMAP/VISI observations. So far, only a few events have been reported about bore events in the southern hemisphere due to limitation of the ground-based airglow measurements. The present work provides insight into the bore structure from large field-of-view space-borne observations.

The following minor revisions are suggested before publication in ACP.

P2 L25: remove "where few observations of bore reported" because you said the same thing in the next sentence.

P2 L29: I cannot follow the meaning "After the variation". Do you mean "validation"?

P3 Event #1: The authors indicate rotation of the bore front. Could you argue that the

round shape front did not expand concentrically from the source?

P7: The authors discussed the undulated wave front in the event #2 from inhomogeneity of propagating speed depending on the duct layer depth. The SABER temperature measurements may suggested the ducting depth. The measurements point of SABER located in a large U area, where larger duct depth is expected. Figure 4b shows a stable layer near the emission height with $\sim$10 km depth or more, which is slightly larger than typical stable layer depth and at least event #1. Also please comment on the cause/mechanism of the horizontal inhomogeneity of temperature gradient with a 1000 km wavelength you suggested, if possible.

---

## Referee Comment (RC2) · Anonymous Referee #2 · 18 Jun 2018

This paper reports two unusual mesospheric bores from an onboard camera on the ISS. While mesospheric bores in the past decades have been described in the literature, this paper is novel in several ways: 1) the reported bores are occurring in southern mid-latitudes (rare), 2) one bore demonstrated a counter clockwise rotation in comparison to clockwise rotation of NH bores (first report), and 3) a large front exhibiting horizontal undulation (first report). Due to these new observations, this paper is worthy of consideration for publication after major revisions. Below are my comments.

Major 1) How is the Brunt-Vaisaila frequency derived from the SABER temperature data? This derivation must depend on an estimation of the derivative. The author should identify which numerical scheme is used and the applied step size to allow proper interpretation of the data. 2) Regarding event 1: In figure 2, it is determined

[Figure]

that there are reflection points separated by ∼5 km in altitude. However, the reflection point (or unstable region) at 95 km appear very narrow and may have a limited impact on the bore. It may be more likely that the lower reflection point is at 90 km. While this should not have much impact on the result stated in the paper, I believe it does deserve a proper discussion. In fact, I recommend the authors perform a calculation of the vertical wavenumber squared, assuming a simple dispersion relation. With that result, the damping impact from the reflection point at 95 km can be estimated. 3) Regarding event 2: Similar to the comment above for event 1. In this case, the narrowly ducted region (3 km) would imply a maximum vertical wavelength of a ducted wave train of ∼6 km. This is less than the anticipated thickness of the airglow layer and one would expect cancellation effects within the emitted airglow. How does that play into the clear observed signature? Again, a simple analysis of the vertical wavenumber may give some indications to whether the statements are within reasonable agreement to the stated conclusions. 4) Line 7.15-16: This sentence needs to be substantiated with an analysis of the vertical wave number.

Minor Recommendations Page 1 Line 1: "...observed by the Visible..." Line 4: "One event was observed over the African..." Line 5-7: Flipping between past and present tense. This should be fixed throughout the paper. Line 7: Change m/sec into m/s. This should be done throughout the paper. Line 7: "...3.5 waves/hour." Line 11: "...undulated with a wavelength of 1000 km" Line 12: FOV is a new acronym. Line 14: "...(SABER) onboard the Thermosphere..." Line 18: "A mesospheric bore is charac- terized by a propagating, and sharp, front in the upper mesosphere." Line 18-19: "The front is often followed by undulations (undular bore) or turbulence (turbulent bore)." Line 19: "Mesospheric bores have been ..."

Line 2.1: "...explanation of a mesospheric bore as a..." Line 2.6: "...Picard (2001) provided a possible explanation of mesospheric bores through critical layer interaction of gravity waves with the mean flow." I just think that "tried" makes it sound as if Dewan and Picard were not successful in their postulation. Line 2.8: "...demonstrated, by

using a numerical simulation, that..." Line 2.9: "...a mesospheric bore from..". This should be fixed throughout the paper. Line 2.10: "...(2010) utilizing ground-based observations." Line 2.10: "generation mechanism" and "origin" are the same, right? Line 2.13: "...than the imagers (FOV)." Line 2.13: FOV was used earlier (Line 1.12) and the acronym definition should be moved to Line 1.12 Line 2.15: replace "wide" with "wider". The ground-based cameras are considered to have a wide FOV, so it would be better to make "wide" as a comparative. Line 2.19: The paper is lacking a clear, strong objective/reasoning to study these waves. I feel that this is a great place where the authors can place their objective. It is pointed out that there are still lot of works to be done. List some of those and then state how your paper addresses these outstanding questions regarding mesospheric bores. Line 2.22: Remove the word "simply". Miller et al. (2015) presented a great work. Instead, try something like "While the focus of the work by Miller et al. (2015) was limited to illustrate the DNB's potential..." Line 2.28: "After the variation,..." I do not know the meaning of this part of the sentence. Line 2.30: The authors should sell the horizontal undulation stronger. While Dewan and Picard discussed undulations, it was vertical undulations and not horizontal. This is, as far as I know, completely new observation and should be highlighted. This has the potential to be a stand out paper for this exact reason. Figure 1: I recommend a full-page figure with the two figures stacked vertically. It will help the reader some of the undulations. Also, Rayleigh is the unit, not label. I suggest writing "Intensity (R)". The last line in the figure caption does not make sense and needs a rewrite. Line 3.6-7: "The spatial resolutions are 13 km along and 12-15 km across the ISS orbit track." Line 3.7: "...ISS is 7.4 km/s, which is significantly higher than..." Line 3.8: Provide some references for the reported bore phase speeds. At a minimum show the references of the papers documenting the lowest and highest bore speed. Line 3.10: "Temperature profiles..." Line 3.11: "...satellite are employed as..." Line 4.8: Missing space between the degree sign and "E". This should be fixed throughout the paper. Line 4.8: "...exactly the same..." Line 4.10: "...front, with an estimated wavelength of 30 km". Line 4.10: How is the horizontal wavelength determined? Line

4.11: "...2,500-3,000 R...". This should be fixed throughout the paper Line 4.15: "...in the western (bright) side as compared to 1,300 R in the eastern (dark) side." Line 4.15: Change westward to eastward. It is stated (correctly) in line 4.13 that the front is moving eastward. Line 4.17: On previous page, it was mentioned that the typical bore speeds are 20 m/s – 100 m/s. I think, the 20 m/s – 100 m/s may be the more extreme limits, whereas 60 m/s -80 m/s are more typical. This should be reflected in the two sentences. Line 4.18: "...increased to seven, implying a wave generation estimated to be 3.5 waves/hour. Line 4.19: "...waves/hour (XXX, XXX, XXX). There are not that many papers detailing wave generation from the leading front, so make references to them here. Line 4.21: It was not expected, it was observed. "...front was observed..." Line 4.26: "...atmospheric tides". There are several tidal modes present. Line 4.26: "...tides make clockwise variations in the northern hemisphere and counter clockwise variations in the southern hemisphere due to the Coriolis acceleration." Line 4.29: "...with the expected background tidal wind variations." The authors should present what these expected tidal wind variations are and provide references. Line 4:29-31: This paragraph could use a rewrite. "Only few reports exist on mesospheric bores in the southern hemispheric midlatitude region (reference(s)) due to the sparse ground-based observation sites. VISI provide the opportunity to study this region and provide more insight into hemispheric differences." Since it is pointed out that only a few studies exist, then provide the list of references. Line 4.32: "TIMED/SABER" made a near-coincident observation..." Line 5.2: "A mesospheric bore is..." Line 5.6: "This mesospheric bore event is likely..." Line 5.12: "A front, characterized by a sharp increase in brightness, elongated W to NW and E to SE was captured..." Line 5.12: I am confused. It was previously stated that a swath was captured every ~3 seconds. In this sentence is seems like to different regions of a swath is captured at different times. Is the swath then comprised of multiple images, and each image is recorded every 3 seconds? If that is the case, then Lines 2.4-5 should be re-written. If not, then this sentence needs more clarification. Line 5.15: "...is the horizontally undulating wave front." This is a key observation of the paper and it should be specific. This is a great observation!

Line 7.2: Since the spatial extend is large, is it possible the evolution of the duct over these scales could be assessed by the previous/following SABER passes? I would be curious to see the N2 analysis for the previous and following SABER passes. Line 7.9: "…bore has never been reported." I agree with this, and I think the authors should consider a new title that captures this. Suggestion: Space-borne mesospheric bore observations by ISS-IMAP/VISI; A first report of an undulating wave front" Line 7.20: "A point-like tropospheric source location of an atmospheric gravity wave can be found by estimating the curvature of the observed wave front from in airglow imagery with the assumption…" Line 8.4: "These results validates the use of VISI for bore studies." Line 8.8: "…tidal backward tidal wind…". I am sure this is a typo. Line 8.12-13. Remove the last sentence.
* * *

---

## Referee Comment (RC3) · Anonymous Referee #3 · 3 Jul 2018

This paper presents new results on mesospheric fronts observed from satellite imaging by using O2(0-0) airglow emission, which are new and interesting, but need substantial revision. Please, see the detailed comments and suggestions just below (the same as in the supplement .pdf file).

The present work describes new mesospheric fronts (bore) observations from the International Space Station (ISS) by using the Visible and near Infrared Spectral Imager (VISI) instrument in the O2 airglow emission.

In general, the manuscript is very well written and presents interesting results but need a deeply revision and include some extra results/discussion. So, a major revision is required.

[Figure]

Just below are presented the major issues found in this version of the manuscript. 1) The title of the manuscript did not reflect the real observations/analysis and I would suggest a little distinct title like this (or similar): "Mesospheric wave front and undulating mesospheric bore observations by ISS-IMAP VISI". The term "Undulating wave front of mesospheric bore" does not reflect the two observed events.

2) In the session 2 (Observations), I would like to suggest to include the Methodology of images analysis (linearization/mapping), image processing and spectral (FFT?) analysis, as well the duct analysis methodology (including equations).

3) The last general suggestion is to include the m2 analysis in order to better discuss the duct in which the mesospheric fronts is propagating. For a deeper discussion, winds from models, or from grand based instrumentation near the region where the fronts were observed, needed to be used. Specific/minor comments: In the "Abstract" and at other part of the text, I suggest to replace m/s by m.s-1, and analogous for other units (e.g., 20°/hour→ 20°.h-1); In the "Introduction", add some recent bore/fronts paper citations, such as: Bageston et al. (2011), Giongo et al. (2018) and Medeiros et al. (2018). On page 2, just before Equation 1, add the word "equations" after ". . .mass and momentum. . .", and replace ";" by ":" in this sentence and at all places where it's appearing ";" instead of ":". After the Equation 1, 'g is gravitational acceleration. . .' would be: 'g is the gravitational acceleration. . .". In the "Observation" session, my doubt is the following: The airglow filter for the O2(0-0) captures only one wavelength or the entire O2(0-0) band, centered at 762 nm? This should be clarified and specified the wavelengths range observed by this filter as well the CCD characteristics, including the quantum efficiency in the observed airglow spectrum, with proper references. Figures should appear just after the respective citation or it's short description. For the first event, Figure 1, the authors mentioned that the "brightness jump" followed by wave structures can be seen around 10°E. This can be better described, since I can see the most intense airglow jump (brightness) around 15°E tilted to 20°E, and two small structures around 5°E. Also, by "measuring" directing in the map one can infer a distance of about 5° between the two small structures (0°-10°E) and the author estimated a horizontal wavelength of about 30 km. So, the method of calculation of the wave parameters should be presented and here a more detailed visualization in the image can be heighted in the map by drawing a straight line between the wave crests. In line 10-12, the authors are showing the airglow intensity in Rayleigh but they did not mention the calibration process of the CCD (sensitivity) and filter/optical system (transmittance) in the instrumentation/observation session. This information is important and need to appear in the instrumentation/observation session. Also, on page 4 (line 16) the bore speed was estimated to be about 100 m.s-1, but it's missing the methodology of wave speed calculation. The number of wave crests and the wave adding rate are dubious since its not possible to identify these characteristics in the image. A very interesting N2 duct appear in Fig 2 (c) but some information on the winds structure would be appreciable, and/or some extra discussion on the kind of duct (thermal or Doppler) by revising some other recent papers (see Bageston et al., 2011 and Giongo et al., 2018). For the Event #2, the same questions/suggestions regarding thee duct as above are valid. On page 6, Figure 4 (c), why the region between 93 and 97 is stable? How the authors can check/prove this stability condition? At the end of page 5 the author said that a small wave structure, parallel to the wave front, is seen at 8°E-0°E. However, the referred region is saturated and the small wave structures can be seen between 10°W and 0°E. To which structure the authors are referring to? There are just a few information on the first panel of Figure 4 (a), and more information could be given. On page 7, in line 1, the word "long" could be put just after the "2,200 km" in order to clarify that this distance is not a wavelength. Line 10-11: The non-uniform bore speed could be easily check by calculating the horizontal wave speed at two or three region of the wave front, that is, in the "Larger U" and at the "Smaller U" regions. Can the author perform this calculation and include the results in the discussions? At least, the Summary can be better written. For example, at line 4: "It is a proof of the VISI validity. . ." could be replaced by: "The results of the present study are a proof. . ." Other improvements in the Summary are expected after the final revision on this paper.

References: Bageston, J. V., Wrasse, C. M., Batista, P. P., Hibbins, R. E., Fritts, D. C., Gobbi, D., and Andrioli, V. F.: Observation of a mesospheric front in a thermal-doppler duct over King George Island, Antarctica, Atmos. Chem. Phys., 11, 12137–12147, https://doi.org/10.5194/acp-11-12137-2011, 2011. Giongo, G. A., Bageston, J. V., Batista, P. P., Wrasse, C. M., Bittencourt, G. D., Paulino, I., Paes Leme, N. M., Fritts, D. C., Janches, D., Hocking, W., and Schuch, N. J.: Mesospheric front observations by the OH airglow imager carried out at Ferraz Station on King George Island, Antarctic Peninsula, in 2011, Ann. Geophys., 36, 253-264, https://doi.org/10.5194/angeo-36-253-2018, 2018. Medeiros, A. F., Paulino, I., Wrasse, C. M., Fechine, J., Takahashi, H., Bageston, J. V., Paulino, A. R., and Buriti, R. A.: Case study of mesospheric front dissipation observed over the northeast of Brazil, Ann. Geophys., 36, 311-319, https://doi.org/10.5194/angeo-36-311-2018, 2018.

Please also note the supplement to this comment:
https://www.atmos-chem-phys-discuss.net/acp-2018-383/acp-2018-383-RC3-supplement.pdf

---

## Author Comment (AC1) · 11 Sep 2018

**acp-2018-383, Author's Reply to the Referees' Comments**

We would like to thank the referees for their detailed and constructive comments. We have revised the manuscript following the comments as is described below. The referee comments are given in green letters. Our replies are given below in black letters with indent. The changes we made in the revised manuscript are given in black bold letters. The page number and line number in our replies indicate those in the revised manuscript, acp-2018-383-revised1-diff.pdf.

Comments from Referee #1

P2 L25: remove "where few observations of bore reported" because you said the same thing in the next sentence.

> We removed it.

P2 L29: I cannot follow the meaning "After the variation". Do you mean "validation"?

> Yes, we meant "validation". We revised the paragraph of Introduction section to described the objective/reasoning of the study more clearly, and removed the sentence.

P3 Event #1: The authors indicate rotation of the bore front. Could you argue that the round shape front did not expand concentrically from the source?

> Thank you for the comment. We added a possible explanation of rotation of the front due to the inhomogeneous phase speed as follows,
>
> **"The difference of the phase speed of the front in the southern side and northern side might be a cause of the rotation. According to equation (1), propagating speed of bore depends on the depth of the ducting layer (U $\propto$ $\sqrt{}$duct depth) and surrounded temperature structure (via g′). If the layer depth is thicker at the southern side, the phase speed is larger there, then, the front rotates counter clockwise."(P6 L14-18).**

P7: The authors discussed the undulated wave front in the event #2 from inhomogeneity of propagating speed depending on the duct layer depth. The SABER temperature measurements may suggested the ducting depth. The measurements point of SABER located in a large U area, where larger duct depth is expected. Figure 4b shows a stable layer near the emission height with ~10 km depth or more, which is slightly larger than typical stable layer depth and at least event #1.

> Thank you for the comment. We agree that the temperature inversion layer in the event #2 has relatively thick depth. We added the temperature profiles from the previous/ following passes to Figure 4. New Figure 4 shows that the stable layer depth from the measurements point located in a large U area is larger than those observed in the previous/following passes. However, it is difficult to directly examine the modulation of layer depth from the SABER measurements. The wave from in the event #2 was modulating with ~ 1,000 km wavelength. As shown in new Figure4a, the east-west

interval of two consecutive SABAR passes is ~ 1,000 km, that is too long to observe the difference of layer depth within one wavelength of the modulation.

Also please comment on the cause/mechanism of the horizontal inhomogeneity of temperature gradient with a 1000 km wavelength you suggested, if possible.

We do not have concrete idea on the cause/mechanism of the horizontal inhomogeneity of temperature gradient or duct depth. However, we can say that the sinusoidal shape of the wave front modulation suggests that the inhomogeneity of duct depth or temperature gradient should have also sinusoidal modulation, and it suggests that the inhomogeneity might be a result of the integration between the stable layer and a wave with horizontal wavelength of 1,000 km. Data of the background structure is very limited in this case. The further discussion on the cause/mechanism of the undulating wave front or possible inhomogeneous duct structure would be difficult.

We revised the discussion section of the event #2, and added following sentences;

**"The bore front modulation in Figure 3 shows sinusoidal shape. It implies that the spatial inhomogeneity in the duct depth or temperature gradient has also sinusoidal modulation. This fact suggests that the modulation in the duct depth or temperature gradient might be a result of the interaction between the stable layer and a wave with horizontal wavelength of 1,000 km."(P8 L12-15).**

Major Comments from Referee #2

1) How is the Brunt-Vaisaila frequency derived from the SABER temperature data? This derivation must depend on an estimation of the derivative. The author should identify which numerical scheme is used and the applied step size to allow proper interpretation of the data.

We added the description of derivation of the Brunt-Vaisaila frequency in "Instruments and Methodology" section (ex-"Observations" section) as follows;

**"The square Brunt-Väisälä frequency can be derived from a temperature profile as**
$$N^2 = g/T \ (\Delta T_z + g/C_p) \quad (2)$$
**where T is the temperature, $\Delta T_z$ is the vertical temperature gradient, and $C_p$ is the specific heat at constant pressure. The square Brunt-Väisälä frequency at an altitude of z is obtained from two continuous SABER temperature data by substituting $T = (T(z_1) + T(z_2))/2$ and $\Delta T_z = (T(z_2) - T(z_2)/(z_2 - z_1)$ into equation (2) where $z = (z_1 + z_2)/2$. The height step of the SABER data around 95 km is ~ 0.4 km."(L4 P17-23).**

2) Regarding event 1: In figure 2, it is determined that there are reflection points separated by ~5 km in altitude. However, the reflection point (or unstable region) at 95 km appear very narrow and may have a limited impact on the bore. It may be more likely that the lower reflection point is at 90 km. While this should not have much impact on the result stated in the paper, I believe it does deserve a proper discussion. In fact, I recommend the authors perform a calculation of the vertical wavenumber squared, assuming a simple dispersion relation. With that result, the damping impact from the reflection point at 95 km can be estimated.

We agree the referee's comment that the unstable region at 95 km may have a limited impact on the bore. We added following sentences to the manuscript;

**"An unstable region around 95 km appears narrow, while an unstable region at 90–91 km is relatively thick. Then, the lower limit of the stable region might be 91 km. In any case, TIMED/SABER data shows the existence of a stable layer near the typical emission peak altitude (95 km) of 762 nm O2 emission."(P6 L4-7).**

The referee recommend us to perform the vertical wavenumber analysis. We agree that the vertical wavenumber analysis would be profitable. However, the vertical wavenumber analysis is difficult or not appropriate for the current case because of following reasons.

If we consider a scenario that a crest of a long-wavelength GW in a duct became sharp and develope into a bore front as simulated by Seyler (2005) and Laughman et al. (2009), the vertical wavenumber analysis for the initial long-wavelength GW would be fruitful to discuss the ducting condition as Yue et al. (2010) did in their analysis. In the current cases, however, the wave parameters, such as phase speed, horizontal wavenumber, which are necessary for the vertical wavenumber analysis, of the primary GW cannot be obtained, since it is near-snapshot imaging from space, and almost no information on the primary GW is available. Therefore, it is impossible to conduct the vertical analysis for the primary GW.

One might consider that the vertical wavenumber analysis should be done for the wave train following the bore front. However, the following wave is thought to be the varicose mode oscillation. According to Dewan and Picard (1998), the wave train following a bore front can be explained with varicose mode oscillation of a stable layer or a duct. In varicose mode, the upper and lower surfaces of the layer oscillate symmetrically about the mid-plane. Medeiros et al., (2005) demonstrated that most of bore events showed that complementary effect suggested by the varicose mode oscillation, and validated the Dewan and Picard (1998)'s explanation. The vertical wavenumber analysis that is usually done for propagating gravity wave should not be applied for the varicose mode oscillation. Therefore, the vertical wavenumber analysis for the wave train following the bore front is not appropriate.

3) Regarding event 2: Similar to the comment above for event 1. In this case, the narrowly ducted region (3 km) would imply a maximum vertical wavelength of a ducted wave train of ~6 km. This is less than the anticipated thickness of the airglow layer and one would expect cancellation effects within the emitted airglow. How does that play into the clear observed signature? Again, a simple analysis of the vertical wavenumber may give some indications to whether the statements are within reasonable agreement to the stated conclusions.

As we mentioned in the reply to the previous comment, the wave train following to the bore front is thought to be varicose mode oscillation. In varicose mode, the upper and lower surfaces of the layer oscillate symmetrically about the mid-plane. Therefore, as far

as the center of the stable layer and the airglow layer is not at exactly same altitude, it is expected that cancellation effects do not affect for the this mode oscillation even when the layer is relatively thin. The vertical wave number analysis that usually do for gravity wave would be not valid for this oscillation. We added description about the varicose mode oscillation in the Introduction section as follows;

**"In varicose mode, the upper and lower surfaces of the layer oscillate symmetrically about the mid-plane. Medeiros et al. (2005) demonstrated that most of bores showed that complementary effect suggested by Dewan and Picard (1998) in airglow responses, and validated the varicose mode oscillation. "(P2 L4-6).**

4) Line 7.15-16: This sentence needs to be substantiated with an analysis of the vertical wave number.

We agree with this referee's comment. However, it is not easy to conduct such an analysis of the vertical wave number with current limited data set. The discussions including the detail analysis about the mechanism of the inhomogeneous duct structure are beyond the scope of this paper. Then, we removed the sentence of Line7.15-16. Instead of that, we added following sentences to the discussion;

**"The bore front modulation in Figure 3 shows sinusoidal shape. It implies that the spatial inhomogeneity in the duct depth or temperature gradient has also sinusoidal modulation. This fact suggests that the modulation in the duct depth or temperature gradient might be a result of the interaction between the stable layer and a wave with horizontal wavelength of 1,000 km."(P8 L12-15).**

Minor Recommendations from Referee #2
Line 7: Change m/sec into m/s. This should be done throughout the paper.

We changed all "m/sec" into "m s^-1". We used "m^-1" format in the revised manuscript following the suggestion from Referee #3.

Line 2.6: "...Picard (2001) provided a possible explanation of mesospheric bores through critical layer interaction of gravity waves with the mean flow." I just think that "tried" makes it sound as if Dewan and Picard were not successful in their postulation.

Thank you for the suggestion. We revised as you suggested.

Line 2.10: "generation mechanism" and "origin" are the same, right?

Yes, we remove "and origin".

Line 12: FOV is a new acronym.
Line 2.13: FOV was used earlier (Line 1.12) and the acronym definition should be moved to Line 1.12

We changed FOV into field-of-view in the abstract. I understand that abstract and main text are separate texts. Then, we remain to use "field-of-view (FOV)" in Line 1.12 because this is first time to use this word in the main text.

Line 2.19: The paper is lacking a clear, strong objective/reasoning to study these waves. I feel that this is a great place where the authors can place their objective. It is pointed out that there are still lot of works to be done. List some of those and then state how your paper addresses these outstanding questions regarding mesospheric bores.

We revised last two paragraphs of Introduction section to show the clear object reasoning of the study as follows;

**"Historically, the study of mesospheric bores have been conducted based on ground-based measurements. Recently, Miller et al. (2015) reported two space-borne observations of mesospheric bore events by Day/Night Band (DNB) onboard the NOAA/NASA Suomi National Polar-orbiting Partnership environmental satellite. Although there are successful observations of mesospheric bores by DNB, a lot of works are left to do on the mesospheric bore study with space-borne imaging. First, the geographical variation on the bore characteristics is not fully understood. The number of ground-based observation site is limited, and the locations of observation site are restricted by land-sea distribution. There are reports of bore observations at low latitudes (e.g., Medeiros et al., 2001; Fechine et al., 2005; Medeiros et al., 2018), northern mid-latitudes (e.g., Taylor et al., 1995; Smith et al., 2003, 2017; Li et al., 2013), and southern high-latitude (e.g., Nielsen et al., 2006; Li et al., 2013; Medeiros et al., 2018), but there is no reports in southern mid-latitudes and northern high-latitudes. Second, the large-scale horizontal structure of bores is unclear. Since ground-based imagers have often observed only a portion of bore's wave front, the typical horizontal spatial scale of mesospheric bore seems to be larger than the imagers' field-of-view (FOV). Previous studies based on ground- based observations have provided limited information on bore's large-scale horizontal structure. Space-borne airglow imaging is a strong tool to study mesospheric bores with global observational coverage and a wider FOV, and can overcome the limitations of ground-based observations.**

**Visible and near Infrared Spectral Imager (VISI) of the Ionosphere, Mesosphere, upper Atmosphere and Plasmasphere mapping mission from the International Space Station (ISS-IMAP mission) is another instrument that has a capability to image**

mesospheric airglow from space with a wide FOV. While Miller et al. (2015) limited **their focus to illustrate the DNB's potential for bore observation, we report two successful bore events from VISI with further detailed analyses to address the two topic, the latitudinal difference of the bore characteristics, and the large horizontal structure. In this paper, we report two mesospheric bores observed by VISI at southern mid-latitudes. The bore of event #1 was captured in two consecutive passes by VISI, thus, the temporal evolution of the structure can be investigated**

**from the difference of two images. The bore showed counter clockwise rotation, while previous studies report clockwise rotation of bore front at northern mid-latitudes (Smith et al., 2003; Li et al., 2013). As event #2, we report a mesospheric bore having a very long wave front exceeding 2,200 km. With a benefit of VISI's wide FOV, a new feature of horizontally undulating wave front was captured. The vertical undulations following bore fronts have been often discussed since Dewan and Picard (1998) explained the following undulation with vertical displacement in varicose mode. However, horizontal undulation of mesospheric bore front has never been reported as far as we know. This paper is the first report of an undulating bore front."(P2 L14-P3 L19).**

Line 2.22: Remove the word "simply". Miller et al. (2015) presented a great work. Instead, try something like "While the focus of the work by Miller et al. (2015) was limited to illustrate the DNB's potential..."

Thank you for the suggestion. We revised as **"While Miller et al. (2015) limited their focus to illustrate the DNB's potential for bore observation, "(P3 L6-7)**.

Line 2.28: "After the variation,..." I do not know the meaning of this part of the sentence.

We revised the paragraph, and removed the part of the sentence.

Line 2.30: The authors should sell the horizontal undulation stronger. While Dewan and Picard discussed undulations, it was vertical undulations and not horizontal. This is, as far as I know, completely new observation and should be highlighted. This has the potential to be a stand out paper for this exact reason.

Thank you for the suggestion. We added sentences to emphasize new observation of horizontal undulation as follows;

**"With a benefit of VISI's wide FOV, a new feature of horizontally undulating wave front was captured. The vertical undulations following bore fronts have been often discussed since Dewan and Picard (1998) explained the following undulation with vertical displacement in varicose mode. However, horizontal undulation of mesospheric bore front has never reported as far as we know. This is a first report of an undulating bore front."(P3 L16-19)**.

Figure 1: I recommend a full-page figure with the two figures stacked vertically. It will help the reader some of the undulations. Also, Rayleigh is the unit, not label. I suggest writing "Intensity (R)". The last line in the figure caption does not make sense and needs a rewrite.

We updated the figure arrangements and the unit labels of Figure 1 and Figure 3. Zoom up images were added to figure 1. The images in Figure 3 were enlarged. The caption of figure 1 was also updated.

Line 3.8: Provide some references for the reported bore phase speeds. At a minimum show the references of the papers documenting the lowest and highest bore speed.

We added the references as follows;

**"The typical phase speed of mesospheric bores is in the range of 60–80 m s$^{-1}$ (e.g., Taylor et al., 1995; Medeiros et al., 2001; Smith et al., 2003; She et al., 2004; Smith et al., 2005; Nielsen et al., 2006; Narayanan et al., 2009; Fechine et al., 2009; Bageston et al., 2011; Li et al., 2013; Giongo et al., 2018). To our knowledge, the highest bore phase speeds reported in the literature are 98 ± 8 m s$^{-1}$ (Brown et al., 2004)."(P4 L7-10)**

Since the possible higher limit of bore space is important in the context, we stated only the typical value and highest value.

Line 4.10: How is the horizontal wavelength determined?

We identified the wave crests in the image by eyes. By measuring the interval of two crests, the horizontal wavelength was determined. We added following sentences;
**"The interval of the crests is ~ 30 km. Then, the wavelength of following waves is ~ 30 km."(P5 L5-6).**
We also updated Figure 1. In the new figure, the identified wave crests are highlighted, and a measure of distance (150 km) was added.

Line 4.17: On previous page, it was mentioned that the typical bore speeds are 20 m/s – 100 m/s. I think, the 20 m/s – 100 m/s may be the more extreme limits, whereas 60 m/s -80 m/s are more typical. This should be reflected in the two sentences.

We agree that more typical value is 60 m/s - 80 m/s. We unified the typical value as 60 m/s - 80 m/s, and revised the sentence in the previous page ("Instrument and Methodology" section) as follows;
**"The typical phase speed of mesospheric bores is in the range of 60–80 m s$^{-1}$ (e.g., Taylor et al., 1995; Medeiros et al., 2001; Smith et al., 2003; She et al., 2004; Smith et al., 2005; Nielsen et al., 2006; Narayanan et al., 2009; Fechine et al., 2009; Bageston et al., 2011; Li et al., 2013; Giongo et al., 2018)."(P4 L7-9)**

Line 4.19: ". . .waves/hour (XXX, XXX, XXX). There are not that many papers detailing wave generation from the leading front, so make references to them here.

We added references as follows; **"that ranged from 1.3 to 8.6 waves h$^{-1}$ (Taylor et al., 1995; Smith et al., 2003; She et al., 2004; Nielsen et al., 2006; Narayanan et al., 2009; Li et al., 2013; Smith et al., 2017)"(P5 L18-19)**. In addition to that, we changed the upper limit of the range from 7 waves h$^{-1}$ to 8.6 waves h$^{-1}$ since we found the larger value in the survey of reference papers (Smith et al., 2017),.

Line 4.29: "...with the expected background tidal wind variations." The authors should present what these expected tidal wind variations are and provide references.

We added references and revised the sentences as follows;
**"Generally, wind in the mesopause region is largely dominated by atmospheric tides and inertia gravity waves, and a majority of those are thought to propagate**

**from lower atmosphere (e.g., Aso and Vincent, 1982; Nakamura et al., 1993). In winter mid-latitude, the semi-diurnal and diurnal tides have significant amplitude (Aso and Vincent, 1982)… ”(P6 L19-22).**

Line 4:29-31: This paragraph could use a rewrite. "Only few reports exist on mesospheric bores in the southern hemispheric midlatitude region (reference(s)) due to the sparse ground-based observation sites. VISI provide the opportunity to study this region and provide more insight into hemispheric differences." Since it is pointed out that only a few studies exist, then provide the list of references.

    We could not find any paper that report mesospheric bore in the southern mid-latitude. Then, We revised as follows;

    **"In the past studies, to our knowledge, there was no report of mesospheric bore at southern mid-latitudes, where the ground-based observation sites are not dense."(P6 L29-30).**

Line 5.12: I am confused. It was previously stated that a swath was captured every ~3 seconds. In this sentence is seems like to different regions of a swath is captured at different times. Is the swath then comprised of multiple images, and each image is recorded every 3 seconds? If that is the case, then Lines 2.4-5 should be re-written. If not, then this sentence needs more clarification.

    We apologize that our manuscript was misleading. A swath was captured every 1.9 seconds, and the 1.9 seconds consist of 1 second exposure time and 0.9 second reading time. We revised the description about the exposure cycle as **"The nominal exposure time is 1.0 s. Since it requires 0.9 s reading time, the exposure cycle is 1.9 s." (P3 L27-28)**. In event 2, ISS flew from NW to SW and each swath, which is perpendicular to the issues orbit track, has NE-SW elongations. All region of a swath were captured at same time. We added arrows to indicate the ISS orbit track direction in Figures 1 and 3.

Line 5.15: ". . .is the horizontally undulating wave front." This is a key observation of the paper and it should be specific. This is a great observation!

    We added a discretion to describe it more specific as follows;

    **"An interesting feature of this event is the undulating wave front. The wave front of the bright jump was not straight. It undulated with a wavelength of ~1,000 km. The crests of the modulation are at (18° W, 27° S) and (6° W, 40° S), where the front advanced forward to the propagating direction (NNE) compared to other portions of the front."(P7 L9-13)**

Line 7.2: Since the spatial extend is large, is it possible the evolution of the duct over these scales could be assessed by the previous/following SABER passes? I would be curious to see the N2 analysis for the previous and following SABER passes.

    We added SABER data of the previous and following passes.

Line 7.9: "...bore has never been reported." I agree with this, and I think the authors should consider a new title that captures this. Suggestion: Space-borne mesospheric bore observations by ISS-IMAP/VISI; A first report of an undulating wave front"

> Thank you for suggestion about the title. We also got a suggestion of new title from referee #3. Following these suggestions, we changed it. The new title is "**Mesospheric bores at southern mid-latitudes observed by ISS-IMAP/VISI; a first report of an undulating wave front**"

Suggestions of correction from Referee #2

Page 1 Line 1: ". . .observed by the Visible. . ."

Line 4: "One event was observed over the African. . ."

Line 5-7: Flipping between past and present tense. This should be fixed throughout the paper.

Line 7: "...3.5 waves/hour."

Line 11: ". . .undulated with a wavelength of 1000 km"

Line 14: ". . .(SABER) onboard the Thermosphere. . ."

Line 18: "A mesospheric bore is characterized by a propagating, and sharp, front in the upper mesosphere."

Line 18-19: "The front is often followed by undulations (undular bore) or turbulence (turbulent bore)."

Line 19: "Mesospheric bores have been . . ."

Line 2.1: "...explanation of a mesospheric bore as a..."

Line 2.8: ". . .demonstrated, by using a numerical simulation, that..."

Line 2.9: "...a mesospheric bore from..". This should be fixed throughout the paper.

Line 2.10: ". . .(2010) utilizing ground-based observations."

Line 2.13: ". . .than the imagers (FOV)."

Line 2.15: replace "wide" with "wider". The ground-based cameras are considered to have a wide FOV, so it would be better to make "wide" as a comparative.

Line 3.6-7: "The spatial resolutions are 13 km along and 12-15 km across the ISS orbit track."

Line 3.7: ". . .ISS is 7.4 km/s, which is significantly higher than. . ."

Line 3.10: "Temperature profiles..."

Line 3.11: "...satellite are employed as…"

Line 4.8: Missing space between the degree sign and "E". This should be fixed throughout the paper.

Line 4.8: "...exactly the same…"

Line 4.10: "...front, with an estimated wavelength of 30 km".

Line 4.11: ". . .2,500-3,000 R. . .". This should be fixed throughout the paper

Line 4.15: ". . .in the western (bright) side as compared to 1,300 R in the eastern (dark) side."

Line 4.15: Change westward to eastward. It is stated (correctly) in line 4.13 that the front is moving eastward.

Line 4.18: ". . .increased to seven, implying a wave generation estimated to be 3.5 waves/hour.

Line 4.21: It was not expected, it was observed. ". . .front was observed. . ."

Line 4.26: ". . .atmospheric tides". There are several tidal modes present.

Line 4.26: ". . .tides make clockwise variations in the northern hemisphere and counter clockwise variations in the southern hemisphere due to the Coriolis acceleration."
Line 4.32: "TIMED/SABER" made a near-coincident observation. . ."
Line 5.2: "A mesospheric bore is. . ."
Line 5.6: "This mesospheric bore event is likely. . ."
Line 5.12: "A front, characterized by a sharp increase in brightness, elongated W to NW and E to SE was captured. . ."
Line 7.20: "A point-like tropospheric source location of an atmospheric gravity wave can be found by estimating the curvature of the observed wave front from in airglow imagery with the assumption. . ."
Line 8.4: "These results validates the use of VISI for bore studies."
Line 8.8: ". . .tidal backward tidal wind. . .". I am sure this is a typo. Line 8.12-13. Remove the last sentence.

Thank you very much for the correction. We corrected the manuscript following these suggestions.

Major Comments from Referee #3
1) The title of the manuscript did not reflect the real observations/analysis and I would suggest a little distinct title like this (or similar): "Mesospheric wave front and undulating mesospheric bore observations by ISS-IMAP VISI". The term "Undulating wave front of mesospheric bore" does not reflect the two observed events.

Thank you for suggestion about the title. We got a suggestion of new title also from referee #3. Following these suggestions, we changed it. The new title is **"Mesospheric bores at southern mid-latitudes observed by ISS-IMAP/VISI; a first report of an undulating wave front".**

2) In the session 2 (Observations), I would like to suggest to include the Methodology of images analysis (linearization/mapping), image processing and spectral (FFT?) analysis, as well the duct analysis methodology (including equations).

We also changed the title of the section into **"Instrumentation and Methodology"**, and revised the section significantly.
We added the description of image mapping as follows;
**"Assuming an ellipsoid with the eccentricity of World Geodetic System 84 (WGS84) and an equatorial axis of wgs84A (6378.137 km) + 95 km as the altitude plane of 95 km, the intersection point between the ellipsoid surface and the line-of-sight of each pixel was calculated. Then, we mapped the airglow intensity to the intersection point, and obtained the two-dimensional airglow image."(P3 L2-5).**

We didn't conduct spectral analysis or specific image processing. We derived the wave parameters from the mapping image. The Figure 1 were updated, and the identified fronts and wave crests are presented.

We added the description of derivation of the Brunt-Vaisaila frequency as follows;
**"The square Brunt-Väisälä frequency can be derived from a temperature profile as**

$$N^2 = g/T \ (\Delta Tz + g/Cp) \qquad (2)$$

**where T is the temperature, $\Delta Tz$ is the vertical temperature gradient, and Cp is the specific heat at constant pressure. The square Brunt-Väisälä frequency at an altitude of z is obtained from two continuous SABER temperature data by substituting T = (T (z1) + T (z2))/2 and $\Delta Tz$ = (T (z2) − T (z2)/(z2 − z1) into equation (2) where z = (z1 + z2)/2. The height step of the SABER data around 95 km is ~ 0.4 km."(L4 P17-23).**

3) The last general suggestion is to include the m2 analysis in order to better discuss the duct in which the mesospheric fronts is propagating. For a deeper discussion, winds from models, or from grand based instrumentation near the region where the fronts were observed, needed to be used.

We agree that the m2 analysis would be profitable. However, it would be difficult to conduct the analysis for the current cases.

If we consider a scenario that a crest of a long-wavelength GW in a duct became sharp and develop into a bore front as simulated by Seyler (2005) and Laughman et al. (2009), m2 analysis for the initial long-wavelength GW would be fruitful to discuss the ducting condition as Yue et al. (2010) did in their analysis. In the current cases, however, the wave parameters, such as phase speed, horizontal wavenumber, which are necessary for m2 analysis, of the primary GW cannot be obtained, since it is near-snapshot imaging from space, and almost no information on the primary GW is available. We might be able to assume background wind from models, even so the wave parameters that is essential for the m2 analysis is not available for the events. Therefore, it is impossible to conduct m2 analysis for the primary GW.

One might consider that m2 analysis should be done for the wave train following the bore front. However, the following wave is thought to be the varicose mode oscillation as we described in the reply to referee #2 Major comment #2 & #3. The m2 analysis that is usually done for propagating gravity wave should not be applied for the varicose mode oscillation.

Specific/minor comments from Referee #3
In the "Abstract" and at other part of the text, I suggest to replace m/s by m.s-1, and analogous for other units (e.g., 20°/hour🔲20°.h-1);

We changed "/s" format into "s^-1" format throughout the manuscript.

In the "Introduction", add some recent bore/fronts paper citations, such as: Bageston et al. (2011), Giongo et al. (2018) and Medeiros et al. (2018).

Thank you for letting us know the recent bore/fronts papers. We added these papers as reference in the introduction section.

On page 2, just before Equation 1, add the word "equations" after "...mass and momentum...", and replace ";" by ":" in this sentence and at all places where it's appearing ";" instead of ":". After the Equation 1, 'g is gravitational acceleration...' would be: 'g is the gravitational acceleration...".

Thank you for the corrections. We corrected them.

In the "Observation" session, my doubt is the following: The airglow filter for the O2(0-0) captures only one wavelength or the entire O2(0-0) band, centered at 762 nm? This should be clarified and specified the wavelengths range observed by this filter as well the CCD characteristics, including the quantum efficiency in the observed airglow spectrum, with proper references.

We significantly revised the observation session. We added the disruptions about the detail of the CCD characteristics and the data processing as follows;

**"VISI has a grism as a disperser, and the spectral resolution ($\lambda/\Delta\lambda$) is ~800. The detector is a back-illuminated, frame transfer CCD (e2V 47-20 AIMO) with 1024 × 1024 pixel format, 13 × 13 μm single pixel size, and 0.92 of quantum efficiency at 630 nm. The column axis of CCD is for space, and the raw axis is for wavelength. On-chip binning is performed along the column (spatial) axis with a nominal binning size of 16 pixel. It has two slit-shape FOVs that are perpendicular to the ISS orbit track and 45∘ forward/backward to nadir. VISI performs continuous line-scan imaging and provides seamless two-dimensional image of airglows. The nominal exposure time is 1.0 s. Since it requires 0.9 s reading time, the exposure cycle is 1.9 s. The main targets of VISI are O2 (0-0) at 762 nm, OH Meinel at 730 nm, and OI at 630 nm. In the nominal operation, the peak(maximum) and background (minimum) counts in the wavelength range around the center (762 nm, 730 nm or 630 nm) ± 6 nm are recorded. By subtracting the background count from the peak count, and multiplying the calibration factor, the airglow intensity is obtained. See Sakanoi et al. (2011) for more detail of the instrumentation."(P3 L22-32).**

Figures should appear just after the respective citation or it's short description.

We apologize that the figure positions were not appropriate. We putted Figure just after the respective citation in the TeX file with a format of ACP template. In the PDF file of the revised manuscript, however, latex run in my environment moved all figures to the end of the sentences. We think the publisher would finalized the manuscript from the TeX file, and the Figure positions would become appropriate.

For the first event, Figure 1, the authors mentioned that the "brightness jump" followed by wave structures can be seen around 10°E. This can be better described, since I can see the most intense airglow jump (brightness) around 15°E tilted to 20°E, and two small structures around 5°E.

Also, by "measuring" directing in the map one can infer a distance of about 5° between the two small structures (0°-10°E) and the author estimated a horizontal wavelength of about 30 km. So, the method of calculation of the wave parameters should be presented and here a more detailed visualization in the image can be heighted in the map by drawing a straight line between the wave crests.

We added following sentences to describe the observation result better;
**"The western (eastern) side of the front is bright (dark), and the boundary is fine. We can see small wave trains whose wavefronts parallel to the front of brightness jump on the western side. The front and wave crests identified in VISI images are indicated with blue lines in Figure 1e, f."(P4 L32-P5 L2)**
**"Another enhancement of airglow brightness can be seen at 15° E–18° E, but the boundary of this enhancement is not sharp, and we will not focus on this enhancement in this study."(P5 L7-8).**

We also update Figure 1 to visualize the wave characteristic more clearly. In the new Figure 1, the bore fronts, wave crests we identified are highlighted with blue lines.
About the horizontal wavelength estimation, we added following sentences;
**"The interval of the crests is ~ 30 km. Then, the wavelength of following waves is ~ 30 km."(P5 L5-6).**

In line 10-12, the authors are showing the airglow intensity in Rayleigh but they did not mention the calibration process of the CCD (sensitivity) and filter/optical system (transmittance) in the instrumentation/observation session. This information is important and need to appear in the instrumentation/observation session.

Yes, it is important information. As we replied above, the observation session was significantly revised. We added the description about optical system and retrieving process of the airglow intensity.

Also, on page 4 (line 16) the bore speed was estimated to be about 100 m.s-1, but it's missing the methodology of wave speed calculation.

We added description about the bore speed estimation as follows;
**"At 50° S, the front moved 620 km eastward in the interval. Thus, if we assume a pure eastward propagation, the bore speed is estimated 100 m /sec s−1. "(P5 L12-14)**.

The number of wave crests and the wave adding rate are dubious since its not possible to identify these characteristics in the image.

In order to display these characteristic more clearly, Figure 1 was updated. The wave crests we identified in the images are indicated by blue lines in Figure 1 (e) and (f).

A very interesting N2 duct appear in Fig 2 (c) but some information on the winds structure would be appreciable, and/or some extra discussion on the kind of duct (thermal or Doppler) by revising some other recent papers (see Bageston et al., 2011 and Giongo et al., 2018).

For the Event #2, the same questions/suggestions regarding thee duct as above are valid.

We agree that a discussion about the background condition including wind information would be preferable. For the current two cases, however, there is no valid wind data because the events were over the ocean, and there was no measurement instrument such as radar or lidar. As far as we know, it is still difficult to assess local wind condition from numerical simulations with several km vertical resolution at upper mesosphere altitudes. Therefore, we did not add extra discussion based on real wind data or numerical simulations. We added descriptions about the possibility of wind contribution to the manuscript as follows;

**"Mesospheric bores are thought to require a channel or region of increased stability, in which a bore propagates (Dewan and Picard, 1998). A mesospheric inversion layer (MIL), a large wind shear, or the combination of both can make such a structure. To assess the background condition of this bore event, both temperature and wind data are ideally needed. For the current case, however, wind data from radar or lidar is not available. Thus, we examine only background temperature structure with TIMED/SABER data. " (P5 L28-33),**

**"As same as event #1, there is no available wind data for this event. We examine only background temperature structure with TIMED/SABER data." (P7 L16-17).**

We also added a description about the importance of wind on the background condition in the introduction section as follows;

**"It has been also demonstrated with simultaneous radar observations that wind has an important role for back- ground condition of mesospheric bore with critical level (Fechine et al., 2009; Giongo et al., 2018)."(P2 L12-13).**

On page 6, Figure 4 (c), why the region between 93 and 97 is stable? How the authors can check/ prove this stability condition?

We use the term "stable layer" with a meaning of a strong stability region (high N2 region) surrounded by low stability or unstable regions (low or negative N2 region). To emphasize this meaning, we added the following description in the manuscript. **"A stable layer, the region of strong stability (high N2) surrounded by low stability or unstable regions (low or negative N2), is recognized at 95-100 km altitude." (P6 L1-3).**

At the end of page 5 the author said that a small wave structure, parallel to the wave front, is seen at 8°E-0°E. However, the referred region is saturated and the small wave structures can be seen between 10°W and 0°E. To which structure the authors are referring to?

We apologize for our mistake. We replaced "8°E-0°E" by "8°W-0°W".

There are just a few information on the first panel of Figure 4 (a), and more information could be given.

We revised the caption of Figure 4 and added the information. New caption is **"SABER measurements obtained from 7 May 22:31 UT to 8 May 01:43 UT. (a) The**

**measurement points at 95 km are indicated by crosses. (b) The altitude profiles of temperature. (c) The altitude profiles of Brunt-Väisälä frequency.".**

On page 7, in line 1, the word "long" could be put just after the "2,200 km" in order to clarify that this distance is not a wavelength.

We added the word.

Line 10-11: The non-uniform bore speed could be easily check by calculating the horizontal wave speed at two or three region of the wave front, that is, in the "Larger U" and at the "Smaller U" regions. Can the author perform this calculation and include the results in the discussions?

VISI observed snapshot of airglow image from space. The horizontal phase speed of the front can not be obtained from the observation for Event 2. Event 1 is a special event that VISI observed a bore front in two consecutive passes. But the front of Event 2 was captured only in one pass, then temporal development was not observed. It is difficult to calculate the horizontal phase speed from the observation.

At least, the Summary can be better written. For example, at line 4: "It is a proof of the VISI validity..." could be replaced by: "The results of the present study are a proof..." Other improvements in the Summary are expected after the final revision on this paper.

We revised the summary paragraph. We hope the paragraph become better. Following the comment from referee #2, the sentence "It is a proof of the VISI validity..." was changed into **"These results validates the use of VISI for bore studies."(P8 L24-25).**

---

## Referee Report (RR1)

Review of **"Mesospheric bore at southern mid-latitudes observed by ISS-IMAP/VISI: a first report of an undulating wave front**" by Yuta Hozumi, Akinori Saito, Takeshi Sakanoi, Atsushi Yamazaki, and Keisuke Hosokawa

This paper describes new mesospheric fronts (bore) observations from the International Space Station (ISS) by using the Visible and near Infrared Spectral Imager (VISI) instrument in the $O_2$ airglow emission.

In this version of the manuscript, the authors deeply revised the paper and now is very well written and presents interesting results with good discussions.

On my opinion the paper is ready for publication.